# Genetic and economic efficiencies of alternative breeding schemes for improvement of local breeds in low-input production systems: The case of the Farta sheep in Northwest Ethiopia

Abiy Shenkut Abebe [1]*, Kefyalew Alemayehu[2], Solomon Gizaw[3]

**1** Department of Animal Science, Debre Tabor University, Debre Tabor, Ethiopia, **2** Department of Animal Science, Bahir Dar University, Bahir Dar, Ethiopia, **3** International Livestock Research Institute (ILRI), Addis Ababa, Ethiopia

* asablome10@gmail.com

## Abstract

Designing and implementing a sound breeding program is essential for sustainably improving livestock productivity. This study evaluated the efficiencies of three breeding schemes for sustainable genetic improvement of indigenous sheep in low-input production systems. The schemes were one-stage selection at six months (Scheme I) or yearling age (Scheme II) and two-stage selections with the first at six months and the second at the yearling age (Scheme III). Each scheme was assessed with three levels of selection proportions (5%, 10% and 20%) and four flock sizes (600, 1200, 1800 and 2400 breeding ewes). Selection responses were simulated using a deterministic approach employed in the SelAction software. For six-month weight, the annual predicted genetic gains ranged from 0.177 to 0.267 kg (Scheme I) and 0.157 to 0.233 kg (Scheme III). For yearling weight, simulated annual genetic gains were 0.268 to 0.399 kg (Scheme II) and 0.265 to 0.398 kg (Scheme III). The expected annual genetic gains for the number of lambs weaned per ewe bred (NLW) and fertility rate were generally small, but the estimates in Schemes II and III were higher compared to Scheme I. The annual economic responses estimated for Schemes I, II, and III ranged from US$0.393 to 0.591, 0.589 to 0.879 and 0.494 to 0.744, respectively. Notably, Scheme II yielded 34% and 16% higher economic returns than Schemes I and III, respectively. The results also revealed that varying the selection proportion significantly influenced the annual selection response, index accuracy, and inbreeding rate. Increasing the flock size had little effect on the genetic progress but significantly reduced the inbreeding rate. Given its genetic and economic benefits alongside operational feasibility, Scheme II, with a 5% selection proportion and a flock size of 1200 breeding ewes, is appropriate for the genetic improvement of indigenous sheep in low-input systems.

**Data availability statement:** All relevant data are within the manuscript and its Supporting Information files.

**Funding:** The author(s) received no specific funding for this work.

**Competing interests:** The authors have declared that no competing interests exist.

## Introduction

Livestock production is a key component of the agriculture sector and has great potential for reducing poverty in developing countries [1]. This is particularly true in Ethiopia, where millions of smallholder farmers and pastoralists depend on livestock to support their livelihood, for instance, in terms of food production and income generation. According to the Central Statistics Agency's most recent report [2], Ethiopia has about 70.3 million, 42.9 million and 52.5 million cattle, sheep and goats, respectively. Given their wide ecological adaptability, sheep are reared by resource-poor farmers and pastoralists in various parts of the country. Ethiopia's northwest highlands are among the country's major sheep-producing areas. Having an estimated population of about 1.2 million heads [2], the Farta sheep breed is one of the local breeds that are widely populated in the alpine and sub-alpine areas of northwest Ethiopia, at elevations ranging from 2,600–3,500 meters above sea level.

Farta sheep are well adapted to a low-input production system, where quality feed shortage and disease incidences are the common challenges [3]. In such a system, the local sheep are entirely managed by smallholder farmers, where breeding decisions are based on their traditional knowledge, with little to no apparent scientific interventions. Besides their promising adaptability to the low-input system, the Farta sheep are among the Ethiopian local sheep breeds that have a high within-breed genetic diversity [4,5]. These sheep serve as multipurpose assets for the smallholder farmers, for instance, providing income, meat, wool and manure [3,6,7]. However, their productivity is generally low, with average weights of 12.37 ± 0.69 kg at six months and 20.08 ± 0.73 kg at a yearling age [8]. This suboptimal performance is primarily linked to the lack of a structured breeding program compatible with the prevailing low-input-based production system.

A conservation-based genetic improvement strategy, such as a community-based or village-based selection breeding program, is reported to be suitable to improve the performance of local sheep reared at the smallholder level [9,10]. Such an approach has also been piloted for some sheep breeds in Ethiopia and has shown promising results, for example, in Menz, Horro and Bonga sheep breeds [11]. However, no such breeding strategy is available for the Farta sheep. In the earlier studies [3,12,13], attempts were made to describe the Farta sheep production system, define breeding objectives and estimate the economic and genetic parameters for multiple traits. Using such information as inputs, the present study simulated alternative breeding schemes for a sustainable genetic improvement of the local sheep in the northwest highlands of Ethiopia, using the Farta sheep breed as a case study. While designing the breeding schemes, the sheep management and breeding practices existing in a low-input sheep production system were taken into account.

## Materials and methods

### Ethics approval statement

Ethical approval is not required because the research does not involve the manipulation of humans or animals.

## Sheep production system

A clear understanding of the existing livestock production system is a critical step in designing breeding programs at the smallholder level [10]. In the northwest highlands of Ethiopia, where the present study is primarily focused on, smallholder farmers' trait preferences, breeding practices and other components of the sheep production system have been well studied [3,14,15]. Previous studies revealed that sheep rearing is an integral part of the crop-livestock mixed farming system in which sheep are raised together with other animal species. The average flock size per smallholder farmer is around 10 sheep. Natural pasture and crop residues are the main feed sources for sheep. The primary distribution areas of Farta sheep in south Gondar zone are shown in Fig 1, highlighting districts with the highest population density. Detailed descriptions of these areas, including sheep breeding practices at the smallholder level, can be found in an earlier study [3].

## Population dynamics and reproduction performances of sheep

A breeding population with discrete generations was assumed. The breeding population is expected to be established by organizing smallholder farmers, who have a common interest in sheep breeding and reside in neighboring villages. The flock size per village typically ranges from 600 to 1200 breeding ewes. Biological parameters reported by Abebe et al. [12] were used to describe the population dynamics of Farta sheep. Accordingly, the survival rate of breeding rams and ewes is about 98%, while the corresponding estimates for lambs from birth to six months and from six months to one year of age are 85.4% and 94.4%, respectively. The lambing rate of ewes is about 76%, with an average litter size of 1.1 lambs. The lambing frequency is about 1.5 per year, or three lambing in two years. Fig 2 illustrates a representative flock of Farta sheep, showing their characteristic features under the smallholder management conditions.

In the present study, the selection was carried out only on males, with no selection criteria imposed on females (apart from breeding soundness). This male-focused approach was adopted because implementing female selection is practically difficult in smallholder systems, where individual farmers typically maintain only a limited number of breeding ewes. Biologically, a single superior ram can service flocks from 3–4 farmers and produce numerous offspring annually, whereas each ewe typically produces only 1–2 lambs per year. Given this difference in reproductive capacity between sexes, sustaining the genetic improvement requires maintaining a larger population of females while selecting fewer higher-quality males, an approach that proves both practical and effective for smallholder production systems. Regarding ram use duration, selected rams are expected to serve for one year with a reasonably controlled mating system.

## Definition of the breeding objective

The principal step when designing a breeding program is the definition of breeding objective, i.e., the choice of traits intended to be genetically improved [16]. We defined the breeding objective based on the trait preferences analysis study reported by Abebe et al. [3]. These authors identified that large body sizes for both breeding rams and ewes, and a good mothering ability for ewes are the most preferred attributes of Farta sheep. It was therefore assumed that the smallholder farmers' breeding goal is to produce Farta sheep with larger body sizes and better mothering ability. To achieve such a breeding goal, six-month weight, yearling weight, the number of lambs weaned per ewe bred (NLW) and ewe fertility rate were considered as the breeding objective traits.

The weights at the six-month and yearling ages are assumed to be the key traits that reflect the body size in male and female sheep. Six-month weight is recorded when lambs are around six months old, often considered the start of the marketing age. Yearling weight is the live weight measured at about 10–12 months of age, at which replacement rams and ewes are expected to start the reproduction cycle. Weights at the six-month and yearling ages have medium to high positive genetic correlations [4,17], indicating that the selection of one of the traits will improve the other too. However, literature sources showed that estimates of genetic gains are far higher when selection is based on yearling weight [18–21]. Thus, the two traits in the current study were assumed to be genetically related but separate traits.

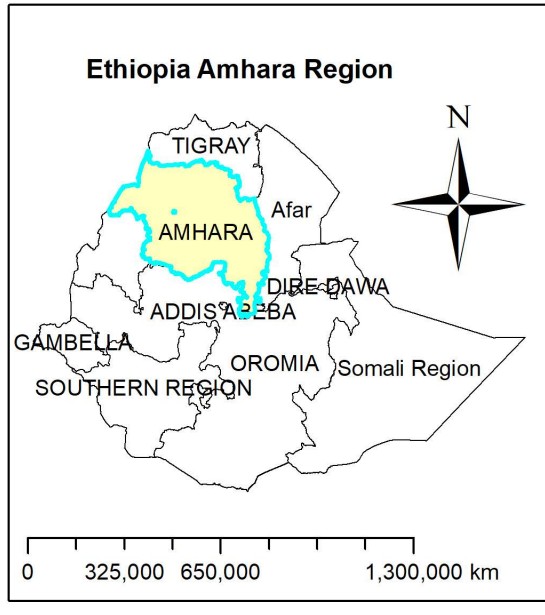

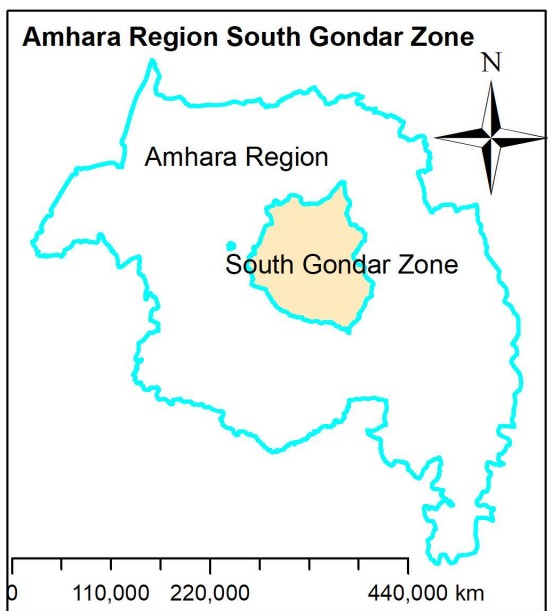

**Estie, Farta and Lay Gayient ditricts of South Gondar Zone**

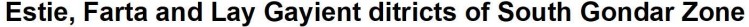

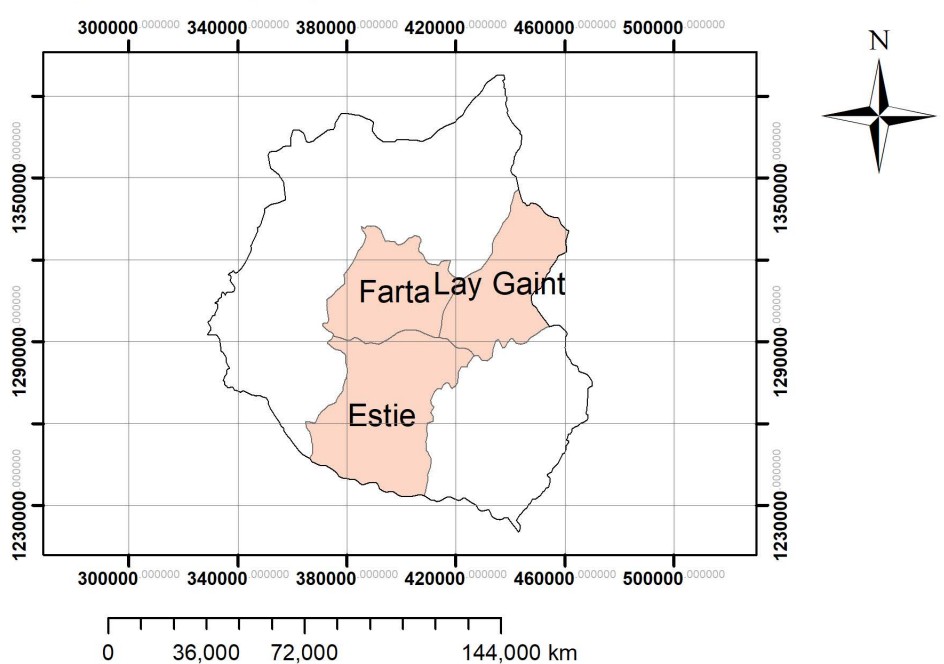

**Fig 1. Major areas in the south Gondar zone where Farta sheep are predominantly found.**

Farmers characterize good mothering ability of breeding ewes in multiple behavioral and reproduction indicators, such as providing better nourishment for their lambs or having more lambs born and surviving up to the weaning age. In this regard, the NLW and fertility rate are presumed to be good indicators of the mothering ability of breeding ewes and,

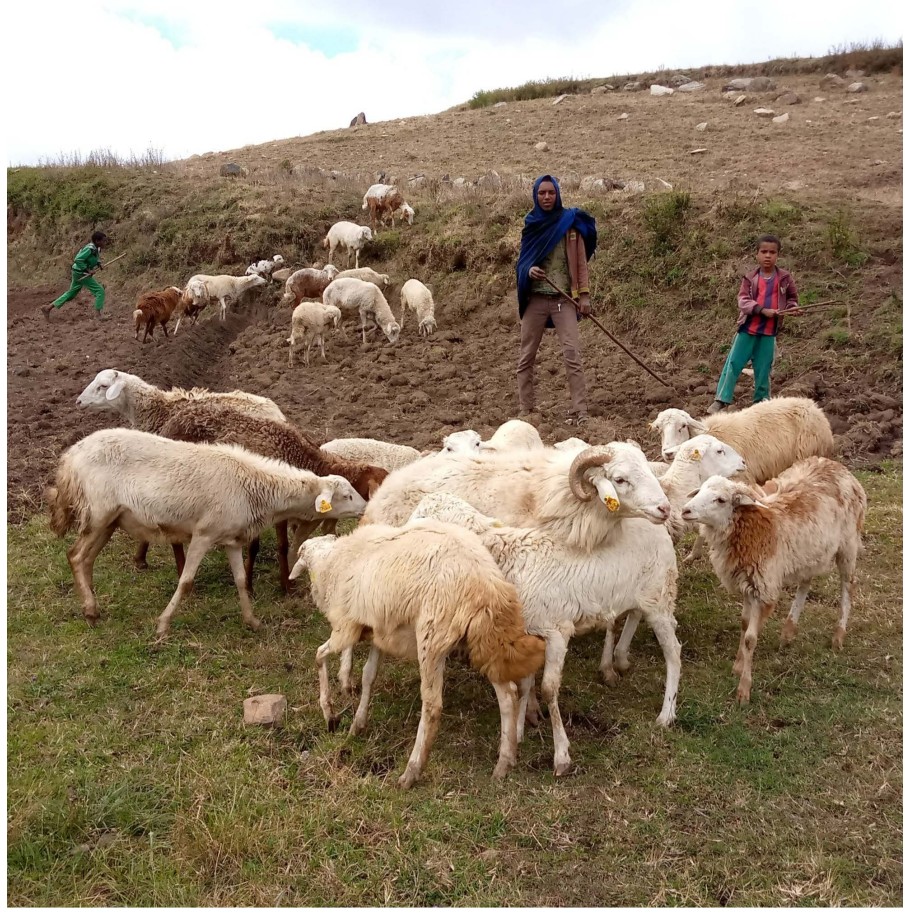

**Fig 2. A representative flock of Farta sheep under the smallholder management conditions.**

hence, play important roles in lamb production. The NLW is a function of litter size at birth and the pre-weaning survival of lambs, while fertility rate refers to the proportion of ewes that gave birth to those bred.

## Alternative breeding schemes

To find an optimal way of achieving the overall breeding objective, three alternative breeding schemes were considered. These schemes take into account sheep production scenarios and breeding practices common in the study areas. The schemes differ in the timing of selection but only slightly vary in the types of traits included in the breeding objective. The details of each Scheme are presented as follows.

## Scheme I

Scheme I involves a one-stage selection at six months or marketing age. The traits incorporated in this scheme are six-month weight, NLW and fertility rate. One of the important benefits of this scheme is that genetic evaluation can be performed at an earlier age. Thus, the resource limitation associated with keeping a large number of selection candidates at the smallholder level may not be a challenge because only selected individuals are maintained until breeding age.

Furthermore, it could also help to address the preference of smallholder farmers who want to sell lambs at an early age, for instance, due to urgent cash needs.

### Scheme II

This scheme assumes a one-stage selection at a yearling or breeding age. The traits are similar to those in Scheme I except that the six-month weight was replaced by yearling weight in Scheme II. This scheme enables direct selection at the yearling age, which could satisfy smallholder farmers' high preference for breeding rams with large body sizes.

### Scheme III

Scheme III involves a two-stage selection process. The first stage had a 50% selection proportion and was applied at 6 months of age. Only those selected in the first stage are eligible for the second-stage selection, which would happen when the animals are yearlings. Traits are similar to Scheme II except that the six-month weight was added as the fourth breeding objective trait. For the first stage, a selection proportion higher or lower than 50% was not included because the results are predictable. For instance, lowering the selection proportion favors the six-month weight at the expense of the yearling weight and vice versa. Scheme III combines selection at the marketing and breeding age, which typically reflects the existing sheep breeding practices in the current study area.

### Evaluation of breeding schemes

The biological and economic efficiencies of the three breeding schemes were evaluated at different sizes of the breeding populations and the proportion of rams selected. For a particular breeding objective trait that appeared in more than one breeding scheme, expected genetic and monetary gains were compared across schemes, but under the same levels of selection proportion and flock size. In addition, the aggregate monetary gains were used to evaluate the economic efficiency of alternative breeding schemes. Furthermore, the effect of flock size and selection proportion was examined by varying one of the factors while keeping the other at the same level.

### Size of the breeding population

To maintain an adequate effective population size, the base breeding unit in a village was set to 600 breeding ewes. Higher flock sizes can be available through cooperation among neighboring villages. Thus, to evaluate the effects of flock size, the number of ewes in the base breeding unit was increased to 1200, 1800 and 2400.

### Ram selection proportion

The average ram-to-ewe mating ratio (1:10) implemented in the traditional sheep breeding practice was considered as a reference [3]. In the current study context, applying a ram-to-ewe mating ratio of 1:10 requires about 60 rams for the 600 breeding ewes, 120 rams for 1200 ewes, and so on. Assuming an equal probability of being male or female at birth and after taking into account the flock dynamics (e.g., mortality, lambing rate, lambing frequency and twinning rate), about 297 male selection candidates are available each year in the base breeding unit containing 600 breeding ewes. Selecting 60 rams from the 297 selection candidates is approximately equal to a selection proportion of 20%. Additionally, efficiencies of the three breeding schemes were evaluated with 5% and 10% selection proportions, which are approximately equal to ram-to-ewe mating ratios of 1:40 and 1:20, respectively. The selection proportions and ram-to-ewe mating ratios for other flock sizes and selection proportions were approximated in the same way as described for a 20% selection proportion, a 1:10 mating ratio, and a flock size of 600 breeding ewes.

### Economic values

According to the selection index theory, the breeding goal is a linear combination of traits desired for improvement, each multiplied by its economic value (EV). In an earlier work by Abebe et al. [12], EVs for a range of traits were estimated

using a bio-economic model based on biological and economic parameters collected in the present study area. However, the EVs reported in the referred source are suitable only for Scheme III because estimates were based on the assumption that about half of the lambs born were sold at six months of age, while the remaining were sold when they were yearlings. Thus, EVs for traits included in Scheme III are US$0.6, 1.46, 0.59 and 0.37 for six-month weight, yearling weight, NLW and fertility rate, respectively.

For Schemes I and II, EVs were modified with the assumption that all non-selected lambs will be sold at six months (Scheme I) or yearling age (Scheme II) using the profit equations developed by Abebe et al. [12]. Accordingly, the recalculated EVs for traits included in the Scheme I are $US 2.19, 0.52 and 0.30 for the six-month weight, NLW and fertility rate, respectively. Whereas for traits in Scheme II, the EVs are $US 2.13, 0.71 and 0.40 for yearling weight, NLW and fertility rate in the order mentioned.

For all the traits, EVs were expressed per ewe per year. While deriving EVs, discounted gene expression effects were not taken into account. Sheep have a relatively short generation interval compared to large animals, such as beef and dairy cattle. Studies have shown that for animal species with a relatively shorter generation interval, accounting for the time delay in trait expression has negligible effects [22,23]. Six-month weight and yearling weight are direct traits for male selection candidates, which are available within a gap of approximately six months. On the other hand, NLW and fertility rate traits are maternal traits, for which breeding values for male selection candidates were estimated from pedigree information. For these reasons, it was assumed that accounting for a discounted genetic expression for this particular study would have a very small effect.

## Genetic and phenotypic parameters

Table 1 shows the genetic and phenotypic correlations, heritability and phenotypic variances assumed for the breeding objective traits. Parameter estimates were obtained from literature sources. For weights at six months and yearling age, the genetic and phenotypic parameters reported for Farta and their cross with Washera sheep were used [13]. However, there are no published literature reports regarding the genetic and phenotypic parameter estimates for NLW and the fertility rate, as well as their correlations with body weight traits for Ethiopian indigenous sheep, or at least for low-input systems in other areas. As a result, all parameter values for the two traits, including their correlations with body weight traits, were based on estimates reported for sheep breeds outside of Ethiopia [24,25]. However, these breeds are not raised in production environments similar to those of the present study areas. Thus, estimates were used with an awareness of their limitations.

## Prediction of selection response and rate of inbreeding

Selection responses and the rate of inbreeding were predicted using a deterministic approach employed in the SelAction software [26]. The software takes into account the decrease in genetic variance due to selection, known as the Bulmer

**Table 1. Genetic and phenotypic parameter estimates of the study traits*.**

| Traits | Six-month weight | Yearling weight | NLW | Fertility rate | Phenotypic variance |
|---|---|---|---|---|---|
| Six-month weight | 0.25 (0.07) | 0.77 (0.12) | 0.20 (0.05) | 0.12 (0.05) | 5.45 |
| Yearling weight | 0.76 (0.01) | 0.3 (0.14) | 0.58 (0.06) | 0.39 (0.07) | 9.37 |
| NLW# | 0.09 (001) | 0.17 (0.02) | 0.13 (0.04) | 0.88 (0.07) | 0.49 |
| Fertility rate | 0.04 (0.01) | 0.09 (0.01) | 0.58 (0.01) | 0.11 (0.04) | 0.12 |

* Genetic correlations (above), phenotypic correlations (below), heritability on the diagonal, standard errors in the parentheses and phenotypic variance on the last column.

# NLW = number of lambs weaned per ewe bred.

effect, while predicting the genetic gain. It also manages the reduction of the intensity of selection linked to the finite population size and the correlation of index values of candidates within the same family.

Information sources included in the indexes for weights at six-month and yearling ages were own performance, pedigree, half-sib and full-sib performances. For NLW and ewe fertility rate, selection was based on pedigree information. For all selection candidates, pedigree information for all breeding objective traits is available at an early age. However, for six-month weight and yearling weight, own performance and sibs' performance records are available at six months and 10–12 months of age, respectively. For breeding Scheme III, the first round of selection was based on pedigree information for all traits, as well as own and sibs' performances for six-month weight, while the second round was based on own and sibs' performances for yearling weight.

It was assumed that data recording on breeding objective traits, including pedigree information, is mandatory. Responses were calculated both in trait and economic units. The total economic response for a particular scheme was derived by summing up the monetary gains of all traits. The selection response per year was computed using an average generation interval of 2.5 years.

## Results and discussion

Optimizing the design of breeding strategies is an important approach commonly applied to identify the most preferred option among alternative schemes, for instance, in terms of the genetic and economic gains and the rate of inbreeding [16,27]. Ideally, a breeding scheme that provides the highest selection response with an acceptable level of inbreeding is the preferred option for long-term animal genetic improvement. While having a breeding scheme with a higher selection response is desirable, it is also equally important to ensure its operational feasibility and compatibility with the existing production system [28].

The present study evaluated the efficiencies of three breeding schemes with varying levels of selection proportion and flock size. All the breeding schemes assumed a single-tier village-based sheep breeding structure. The results highlighted that a reasonable amount of genetic progress can be achieved in the traits of Ethiopian indigenous sheep using selective breeding programs. More specifically, Scheme II offered higher economic benefits than the other schemes. Additionally, it was observed that selection proportion largely influenced the genetic and economic responses, while both the selection proportion and flock size affected the rate of inbreeding.

### Expected genetic gains for the breeding objective traits

Annual genetic gains and the accuracy of indexes, predicted for traits in the studied schemes, are presented in Table 2. For the six-month weight, the genetic gains estimated using Schemes I and III varied from 0.177 to 0.267 kg/year and 0.157 to 0.233 kg/year, respectively. The results indicate that expected genetic gains in Scheme I were 11–13% higher than those predicted in Scheme III. Overall, the selection responses predicted for the six-month weight in the current study fall within the range of earlier findings. For instance, Dagnew et al. [19] obtained an annual genetic gain of 0.171 kg/year for the six-month weight, while Gizaw et al. [21] reported estimates ranging from 0.119 to 0.285 kg/year for other sheep breeds of Ethiopia.

The genetic gains in yearling weight predicted using Schemes II and III ranged from 0.268 to 0.399 kg/year and 0.265 to 0.398 kg/year, respectively (Table 2). The results imply that genetic gains in the yearling weight are nearly the same between the two schemes. Compared with literature estimates, the annual genetic gains in yearling weight obtained in the present study were relatively low. For instance, Gizaw et al. [18] reported higher genetic gains (0.49 to 0.70 kg/year) for the yearling weight of the Menz sheep breed, using similar selection proportions to the current study. Similarly, Mirkena et al. [20] estimated genetic gains of 0.440 to 0.94 kg/year for the same trait in four sheep breeds, albeit with lower index accuracies (0.26 to 0.30) than those in the present study (0.60 to 0.62).

Disparities in the estimates of annual genetic gains for the yearling weight might be due to performance variations among sheep breeds, as relatively larger additive genetic variances were used in the referenced literature sources.

Variations also exist in the types of breeding objective traits and information sources included in the selection index between the present study and the cited sources, which are likely to contribute to differences in the estimates of genetic gains.

The expected annual genetic gains for NLW and fertility rate were generally small, though larger in Schemes II and III than in Scheme I (Table 2). For instance, with a 5% selection proportion and a flock size of 2400 ewes, the maximum genetic gains predicted in Schemes II and III were around 0.035 and 0.033 lambs/year in NLW and about 0.011 and 0.01 improvements in the fertility rate, respectively. In contrast, under the same selection proportion and flock size, the expected genetic responses in Scheme I were about 0.012 lambs/year for NLW and 0.003 for fertility rate. Since the selection was carried out only on males and NLW and fertility rate are maternal traits, the genetic prediction relied on parental pedigree performance and correlated responses from genetically related traits. Given such circumstances, a higher genetic gain is unlikely to be achieved in NLW and the fertility rate. Additionally, the low heritability estimates for these traits indicate limited genetic variation [24,29], further explaining the slow genetic progress.

**Table 2. Genetic gains per year and index accuracies for the three breeding schemes.**

| Traits in the breeding schemes (units) | Ram selection proportion | | | | | | | | | | | |
|---|---|---|---|---|---|---|---|---|---|---|---|---|
| | P=0.05 | | | | P=0.1 | | | | P=0.2 | | | |
| | Number of breeding ewes | | | | Number of breeding ewes | | | | Number of breeding ewes | | | |
| | 600 | 1200 | 1800 | 2400 | 600 | 1200 | 1800 | 2400 | 600 | 1200 | 1800 | 2400 |
| Scheme I | | | | | | | | | | | | |
| Six-month weight (kg) | 0.262 | 0.265 | 0.266 | 0.267 | 0.223 | 0.224 | 0.224 | 0.224 | 0.177 | 0.178 | 0.178 | 0.178 |
| NLW (lambs)* | 0.011 | 0.012 | 0.012 | 0.012 | 0.010 | 0.010 | 0.010 | 0.010 | 0.008 | 0.008 | 0.008 | 0.008 |
| Fertility rate | 0.003 | 0.003 | 0.003 | 0.003 | 0.003 | 0.003 | 0.003 | 0.003 | 0.002 | 0.002 | 0.002 | 0.002 |
| Accuracy of index | 0.586 | 0.586 | 0.586 | 0.586 | 0.573 | 0.573 | 0.573 | 0.573 | 0.560 | 0.560 | 0.560 | 0.560 |
| Scheme II | | | | | | | | | | | | |
| Yearling weight (kg) | 0.393 | 0.397 | 0.398 | 0.399 | 0.335 | 0.337 | 0.337 | 0.338 | 0.268 | 0.268 | 0.269 | 0.269 |
| NLW (lambs) | 0.034 | 0.035 | 0.035 | 0.035 | 0.029 | 0.030 | 0.030 | 0.030 | 0.023 | 0.024 | 0.024 | 0.024 |
| Fertility rate | 0.010 | 0.011 | 0.011 | 0.011 | 0.009 | 0.009 | 0.009 | 0.009 | 0.007 | 0.007 | 0.007 | 0.007 |
| Accuracy of index | 0.615 | 0.615 | 0.615 | 0.615 | 0.605 | 0.605 | 0.605 | 0.605 | 0.595 | 0.595 | 0.595 | 0.595 |
| Scheme III | | | | | | | | | | | | |
| Stage one selection | | | | | | | | | | | | |
| Six-month weight (kg) | 0.098 | 0.099 | 0.099 | 0.099 | 0.097 | 0.097 | 0.097 | 0.097 | 0.095 | 0.096 | 0.096 | 0.096 |
| Yearling weight (kg) | 0.123 | 0.124 | 0.124 | 0.124 | 0.122 | 0.122 | 0.123 | 0.123 | 0.122 | 0.122 | 0.122 | 0.122 |
| NLW (lambs) | 0.006 | 0.006 | 0.006 | 0.006 | 0.006 | 0.006 | 0.006 | 0.006 | 0.007 | 0.007 | 0.007 | 0.007 |
| Fertility rate | 0.002 | 0.002 | 0.002 | 0.002 | 0.002 | 0.002 | 0.002 | 0.002 | 0.002 | 0.002 | 0.002 | 0.002 |
| Accuracy of index | 0.520 | 0.520 | 0.520 | 0.520 | 0.511 | 0.511 | 0.511 | 0.511 | 0.504 | 0.504 | 0.504 | 0.504 |
| Stage two selection | | | | | | | | | | | | |
| Six-month weight (kg) | 0.229 | 0.232 | 0.232 | 0.233 | 0.196 | 0.196 | 0.197 | 0.197 | 0.157 | 0.157 | 0.158 | 0.158 |
| Yearling weight (kg) | 0.392 | 0.396 | 0.397 | 0.398 | 0.334 | 0.336 | 0.336 | 0.336 | 0.265 | 0.265 | 0.266 | 0.266 |
| NLW (lambs) | 0.032 | 0.032 | 0.032 | 0.033 | 0.027 | 0.027 | 0.027 | 0.027 | 0.021 | 0.021 | 0.021 | 0.021 |
| Fertility rate | 0.010 | 0.010 | 0.010 | 0.010 | 0.008 | 0.008 | 0.008 | 0.008 | 0.006 | 0.006 | 0.006 | 0.006 |
| Accuracy of index | 0.617 | 0.617 | 0.617 | 0.617 | 0.607 | 0.607 | 0.607 | 0.607 | 0.597 | 0.597 | 0.597 | 0.597 |
| Correlation of accuracies# | 0.843 | 0.843 | 0.843 | 0.843 | 0.842 | 0.842 | 0.842 | 0.842 | 0.844 | 0.844 | 0.844 | 0.844 |

*NLW=number of lambs weaned per ewe bred.

# Correlation of accuracies=calculated as the ratio of the accuracy of stage one to stage two, following Rutten et al. [26].

Published estimates of genetic gains for ewe fertility rate are not available for comparison with the present results. For the NLW trait, however, Gizaw et al. [18] reported lower genetic gains (0.003 to 0.004 lambs/year) in another Ethiopian indigenous sheep. Such differences may reflect breed-specific variation in lamb survival and litter sizes, as disease prevalence and husbandry practices vary across the different areas of Ethiopia. Furthermore, genetic parameter estimates of NLW in the present study were used from other sheep breeds, which may have additionally contributed to the observed variation.

## Expected economic responses of breeding schemes

Table 3 summarizes the annual economic returns estimated for the breeding goal traits in the three breeding schemes. Across the different selection proportions and flock sizes, the total economic responses predicted for Schemes I, II and III ranged from US$0.393 to 0.591, US$0.589 to 0.879 and US$0.494 to 0.744, respectively. The results imply that Scheme II yields about 16% and 34% higher economic returns than Schemes III and I, respectively, while Scheme III offers about 20% higher economic returns than Scheme I. It was also observed that the genetic improvement in six-month and yearling weights accounted for about 99% and 97% of the total economic gains in Schemes I and II, respectively. The high

**Table 3. Economic response per year (US$) for breeding goal traits in the three breeding schemes.**

| Traits in the breeding schemes | Ram selection proportion | | | | | | | | | | | |
| --- | --- | --- | --- | --- | --- | --- | --- | --- | --- | --- | --- | --- |
| | P=0.05 | | | | P=0.10 | | | | P=0.20 | | | |
| | Number of breeding ewes | | | | Number of breeding ewes | | | | Number of breeding ewes | | | |
| | 600 | 1200 | 1800 | 2400 | 600 | 1200 | 1800 | 2400 | 600 | 1200 | 1800 | 2400 |
| Expected economic response per year (US$) | | | | | | | | | | | | |
| Scheme I | | | | | | | | | | | | |
| Six-month weight | 0.574 | 0.581 | 0.583 | 0.584 | 0.488 | 0.490 | 0.491 | 0.492 | 0.388 | 0.388 | 0.389 | 0.389 |
| NLW* | 0.006 | 0.006 | 0.006 | 0.006 | 0.005 | 0.005 | 0.005 | 0.005 | 0.004 | 0.004 | 0.004 | 0.004 |
| Fertility rate | 0.001 | 0.001 | 0.001 | 0.001 | 0.001 | 0.001 | 0.001 | 0.001 | 0.001 | 0.001 | 0.001 | 0.001 |
| Total economic response# | 0.581 | 0.588 | 0.590 | 0.591 | 0.494 | 0.496 | 0.497 | 0.498 | 0.393 | 0.393 | 0.394 | 0.394 |
| Scheme II | | | | | | | | | | | | |
| Yearling weight | 0.837 | 0.846 | 0.849 | 0.850 | 0.713 | 0.717 | 0.718 | 0.719 | 0.570 | 0.572 | 0.573 | 0.573 |
| NLW | 0.024 | 0.025 | 0.025 | 0.025 | 0.021 | 0.021 | 0.021 | 0.021 | 0.016 | 0.017 | 0.017 | 0.017 |
| Fertility rate | 0.004 | 0.004 | 0.004 | 0.004 | 0.004 | 0.004 | 0.004 | 0.004 | 0.003 | 0.003 | 0.003 | 0.003 |
| Total economic response | 0.864 | 0.875 | 0.878 | 0.879 | 0.738 | 0.742 | 0.743 | 0.744 | 0.589 | 0.592 | 0.593 | 0.593 |
| Scheme III | | | | | | | | | | | | |
| Stage one selection | | | | | | | | | | | | |
| Six-month weight | 0.059 | 0.059 | 0.060 | 0.060 | 0.058 | 0.058 | 0.058 | 0.058 | 0.057 | 0.057 | 0.057 | 0.058 |
| Yearling weight | 0.179 | 0.180 | 0.181 | 0.181 | 0.178 | 0.179 | 0.179 | 0.179 | 0.178 | 0.178 | 0.178 | 0.178 |
| NLW | 0.004 | 0.004 | 0.004 | 0.004 | 0.004 | 0.004 | 0.004 | 0.004 | 0.004 | 0.004 | 0.004 | 0.004 |
| Fertility rate | 0.001 | 0.001 | 0.001 | 0.001 | 0.001 | 0.001 | 0.001 | 0.001 | 0.001 | 0.001 | 0.001 | 0.001 |
| Stage two selection | | | | | | | | | | | | |
| Six-month weight | 0.138 | 0.139 | 0.140 | 0.140 | 0.117 | 0.118 | 0.118 | 0.118 | 0.094 | 0.094 | 0.094 | 0.094 |
| Yearling weight | 0.572 | 0.578 | 0.580 | 0.581 | 0.488 | 0.490 | 0.491 | 0.491 | 0.386 | 0.388 | 0.388 | 0.388 |
| NLW | 0.019 | 0.019 | 0.019 | 0.019 | 0.016 | 0.016 | 0.016 | 0.016 | 0.012 | 0.012 | 0.012 | 0.012 |
| Fertility rate | 0.004 | 0.004 | 0.004 | 0.004 | 0.003 | 0.003 | 0.003 | 0.003 | 0.002 | 0.002 | 0.002 | 0.002 |
| Total economic response | 0.733 | 0.740 | 0.743 | 0.744 | 0.624 | 0.627 | 0.628 | 0.628 | 0.494 | 0.496 | 0.496 | 0.496 |

* NLW = number of lambs weaned per ewe bred.

# Total economic response = the sum of the economic gains in the breeding goal traits.

economic contributions of weights at the six-month and yearlings indicate that the two traits are vital to maximizing the profitability of sheep breeding at the smallholder level.

Body weight traits have relatively high heritability and are easily measurable, implying that selection for these traits is comparatively easier than for other traits. However, long-term selection for larger body sizes in low-input systems may pose challenges. For instance, breeding ewes might need to allocate a higher proportion of their feed intake to meet maintenance requirements. In this regard, setting an optimal improvement threshold for body size is important.

## Effects of selection proportion and flock size

Varying the selection proportion had a significant effect on the selection response. It was observed that keeping all the other scenarios constant, the annual selection responses predicted using a 5% selection proportion were higher by up to 50% than those under a 20% selection proportion (Tables 2 and 3). Given their direct relationship, an increased selection intensity is expected to result in a higher selection response. However, applying a strong selection intensity in low-input sheep production systems can be challenging due to the small number of breeding females per smallholder farm. One of the viable options to overcome such a challenge is to organize the participant farmers based on communal resources, such as grazing land and watering points [20,30].

Another approach is the use of artificial insemination using fresh semen from a few genetically superior rams, which has shown promise for accelerating breeding in Ethiopian indigenous sheep [31]. Given such enabling situations, the high selection response linked to the strong ram selection intensity can be exploited for indigenous sheep genetic improvement at the smallholder level in the northwest highlands of Ethiopia.

Table 4 provides an overview of the rate of inbreeding per generation. SelAction software predicts the rate of inbreeding for one-stage selection only. It was assumed that inbreeding in one-stage (Schemes I and II) and two-stage (Scheme III) selections are comparable because all schemes were evaluated under the same selection proportion and flock size. The results revealed that applying a 5% selection proportion with a flock size of 600 ewes led to an inbreeding rate exceeding the 1% threshold typically deemed acceptable in livestock genetic improvement programs. However, increasing the flock size to 1200 breeding ewes (while keeping the 5% selection proportion) significantly reduced the rate of inbreeding per generation. This reduction occurs because a higher flock size increases the number of selected parents, thereby decreasing genetic similarity among mating pairs.

Table 4. The effects of selection proportion and flock size on the rate of inbreeding per generation.

| Selection proportions | Flock size | Rate of inbreeding per generation (%) | |
|---|---|---|---|
| | | Scheme I | Scheme II |
| 5% | 600 | 1.320 | 1.352 |
| | 1200 | 0.688 | 0.706 |
| | 1800 | 0.464 | 0.476 |
| | 2400 | 0.350 | 0.359 |
| 10% | 600 | 0.540 | 0.568 |
| | 1200 | 0.273 | 0.288 |
| | 1800 | 0.188 | 0.192 |
| | 2400 | 0.141 | 0.146 |
| 20% | 600 | 0.220 | 0.233 |
| | 1200 | 0.110 | 0.118 |
| | 1800 | 0.074 | 0.078 |
| | 2400 | 0.055 | 0.059 |

Alternatively, lowering the selection proportion minimized the level of inbreeding across all flock sizes but also substantially reduced the genetic gain. Similarly to the present study, Bett et al. [16] and Gizaw et al. [18] reported higher effects of selection proportion and flock size on the rate of inbreeding. Overall, balancing selection proportion and the flock size is crucial to minimize inbreeding while keeping genetic progress in long-term breeding programs.

In contrast to the selection proportion, varying the flock size showed negligible effects on both the annual selection response and index accuracies. As shown in Table 3, Scheme II yielded an economic response of about US$ 0.864 per year with a 5% selection proportion and a flock size of 600 breeding ewes. Nevertheless, quadrupling the flock size with the same level of selection proportion increased annual economic return only by 1.7%. In all the schemes, the effect of flock size on the economic gain diminished further when the selection proportions were relaxed to 10% and 20%. It is worth mentioning that having a large flock size is important to get a large number of improved rams that can be used to accelerate genetic improvement at the breed level. However, considering a flock size higher than 1200 breeding ewes could be practically difficult due to the organizational challenges associated with poor infrastructure and resource limitations inherent to low-input production systems.

## Operational feasibility of breeding schemes

In general, all the breeding schemes rendered positive genetic and economic gains. This indicates that implementing either of the three schemes would result in genetic improvements in the studied traits. However, alternative schemes differ in their efficiencies and organizational and technical complexity. Scheme I is operationally more feasible than the others because data can be available at six months of age before lambs are sold, but it offers a relatively low economic response. Scheme III fits well with the existing sheep breeding practice because it considers different lamb-selling ages practiced by farmers. However, Scheme III seems technically more difficult to implement at smallholder levels than other schemes because selection needs to be implemented in two stages.

Although Scheme II requires maintaining all the selection candidates up to a yearling age, which demands more resources such as feed, it is operationally easier than Scheme III. It also provides a relatively higher economic response than both Schemes I and III. Considering its genetic and economic benefits alongside operational feasibility, Scheme II is assumed to be a more preferred strategy for the genetic improvement of sheep in low-input production systems.

Smallholder farmers often keep a relatively small number of sheep, which is a common coping mechanism for resource limitations under a low-input system. In other words, resource limitation could become a challenge if the flock size per farmer increases. Implementing Scheme II should aim to improve the genetic performance of the small number of sheep available at the individual farmer level by collaborating with other farmers to create a large breeding population. However, improved genotypes require quality feed and good health management. Thus, farmers should be aware of these necessities to achieve sustainable genetic improvement. Additionally, considering a breeding population larger than 1200 breeding ewes may also be organizationally challenging, particularly for tasks related to data recording and organizing a large number of farmers. Therefore, applying Scheme II with a selection proportion of 5% and a breeding population of 1200 ewes is optimal in terms of annual genetic and economic gains with an acceptable level of inbreeding.

## Conclusion

The current study evaluated the efficiencies of three breeding schemes for optimizing the genetic improvement strategies in low-input sheep production systems. The results highlighted that balancing selection proportion with flock size is vital for sustainable genetic improvement. Scheme I, which involves a one-stage selection at six months of age, can easily be implemented at the smallholder level, but expected genetic and economic responses are relatively low. The annual economic gain in Scheme II is higher than in Schemes I and III. Again, Scheme II seems operationally more feasible than Scheme III because all selection candidates are evaluated once at yearling age. Overall, Scheme II, using a 5% selection proportion and a flock size of 1200 breeding ewes, would be a better genetic improvement strategy for low-input-based

sheep breeding areas in general and the northwest highlands of Ethiopia in particular. In addition, implementing Scheme II could be a good entry point to develop breeding structures involving more tiers over time. This can be achieved by using elite breeder farmers to produce and utilize superior breeding stock and fellow breeder farmers to expand the genetic improvement at the breed level.

## Supporting information

**S1 File. Input data for simulation in SelAction 2.1 program.** It is an Excel file (XLSX) containing all the input data for Schemes I, II and III.
(XLSX)

## Acknowledgments

The first author would like to acknowledge smallholder farmers in the South Gondar Zone for their unwavering support and invaluable feedback during the formulation of the alternative breeding schemes addressed in this study.

## Author contributions

**Conceptualization:** Abiy Shenkut Abebe, Kefyalew Alemayehu, Solomon Gizaw.

**Data curation:** Abiy Shenkut Abebe.

**Formal analysis:** Abiy Shenkut Abebe.

**Investigation:** Abiy Shenkut Abebe, Kefyalew Alemayehu.

**Methodology:** Abiy Shenkut Abebe.

**Supervision:** Kefyalew Alemayehu, Solomon Gizaw.

**Validation:** Kefyalew Alemayehu, Solomon Gizaw.

**Writing – original draft:** Abiy Shenkut Abebe, Solomon Gizaw.

**Writing – review & editing:** Kefyalew Alemayehu.

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
