## [Decision Letter · Decision Letter 0]

26 Jun 2025

Dear Dr. Abebe,

Thank you for submitting your manuscript to PLOS ONE. After careful consideration, we feel that it has merit but does not fully meet PLOS ONE’s publication criteria as it currently stands. Therefore, we invite you to submit a revised version of the manuscript that addresses the points raised during the review process.

We look forward to receiving your revised manuscript.

Kind regards,

Dawit Tesfaye

Academic Editor

PLOS ONE

Journal Requirements:

Additional Editor Comments :

The mansucript is revised by experts in the field and they raised a significant concern on the technical aspects and presentation of the data. The mansucript need a major revision prior to condideration for further review and decision.

Reviewers' comments:

Reviewer's Responses to Questions

**Comments to the Author**

1. Is the manuscript technically sound, and do the data support the conclusions?

Reviewer #1: Yes

Reviewer #2: No

2. Has the statistical analysis been performed appropriately and rigorously?

Reviewer #1: Yes

Reviewer #2: No

3. Have the authors made all data underlying the findings in their manuscript fully available?

Reviewer #1: Yes

Reviewer #2: Yes

4. Is the manuscript presented in an intelligible fashion and written in standard English?

Reviewer #1: Yes

Reviewer #2: No

Reviewer #1: Introduction

Emphasize why Farta sheep is chosen by mentioning their socio- economic or ecological importance

It should be explained why selection is performed on male?

Material and Methods

Consider adding a country map highlighting the study area, and also add an image of Farta Sheep breed

Minor comments

There is some repetition and minor grammatical issues

Strength

The manuscript is well structured with a logical flow of ideas. Citations are properly used to support key points. Breeding schemes are clearly explained with practical considerations. It sets the context by highlighting the importance of livestock in Ethiopia and provides adequate background on Farta sheep breed.

Reviewer #2: materials and methods too elaborate and not information

method of analysis not clearly explained

Table 1. Genetic and phenotypic parameter estimates of the study traits need standard error and only few traits have been studied and the reason for the same

**Do you want your identity to be public for this peer review?** For information about this choice, including consent withdrawal, please see our Privacy Policy

Reviewer #1: **Yes: ** Genet Dejene Dibaba

Reviewer #2: No

---

## [Author Response · Author response to Decision Letter 1]

2 Aug 2025

Dear Editor and Reviewers,

We sincerely appreciate your comments and suggestions on our manuscript. We have carefully addressed your valuable insights in our revisions and hope they now meet your expectations. Below, we provide detailed responses to each point.

I. Comments from the Academic Editor

Journal Requirements:

1. Please ensure that your manuscript meets PLOS ONE's style requirements, including those for file naming. The PLOS ONE style templates can be found at https://journals.plos.org/plos

one/s/file?id=wjVg/PLOSOne_formatting_sample_main_body.pdf and https://journals.p

los.org/plosone/s/file?id=ba62/PLOSOne_formatting_sample_title_authors_affiliations.pdf

Authors’ Response:

We sincerely appreciate the valuable reminder regarding this important issue. The authors have carefully prepared the manuscript in strict accordance with PLOS ONE's guidelines and have made every effort to ensure full compliance with the journal's style requirements.

Authors’ Response:

We appreciate the Editor's comment regarding our data availability statement. As noted, we have included all input parameters as supplementary information to ensure full transparency. We confirm that all data supporting the findings of this study are available within the manuscript and its supporting information files.

Additional Editor Comments:

The mansucript is revised by experts in the field and they raised a significant concern on the technical aspects and presentation of the data. The mansucript need a major revision prior to condideration for further review and decision.

Authors’ Response:

We sincerely thank the reviewers' valuable comments & questions, and have carefully addressed each point in our revision. Below, we provide detailed responses to all concerns raised.

II. Comments and questions from reviewer #1

1. Introduction

1.1. Emphasize why Farta sheep is chosen by mentioning their socio- economic or ecological importance

Authors’ Responses:

Dear reviewer #1,

Thank you for raising this important issue. As highlighted in the Introduction section, we have provided detailed descriptions of Farta sheep, including:

✓ Population size at the breed level

✓ Adaptation to alpine and subalpine ecologies (thriving in extremely cold climates)

✓ Resilience in low-input systems, where feed shortages and disease outbreaks are common challenges

✓ High within-breed genetic diversity, indicating strong potential for genetic improvement through selective breeding

✓ Socio-economic importance, as they provide income, meat, wool, and manure for local communities

Despite their ecological and socioeconomic significance, Farta sheep exhibit relatively low average growth performance, and no prior attempts have been made to address this limitation. This gap motivated our study to develop alternative breeding schemes for this breed.

We believe these points adequately justify the necessity of a tailored breeding program for the genetic improvement of Farta sheep.

1.2. It should be explained why selection is performed on male?

Authors’ Responses:

Dear reviewer #1,

We addressed this question in the material and method section of the manuscript, as it is a methodological issue.

We assumed selection only on males, with no selection on females (except for breeding soundness), because it’s practically difficult to apply selection on females due to the small number of breeding ewes available per individual farmer. To sustain the genetic improvement, all female offspring from selected rams should be incorporated into the breeding population. The breeding strategy requires more number of females compared to a relatively fewer males because a single genetically superior ram can service flocks from 3-4 farmers and produce multiple offspring annually, while each ewe produces only 1-2 lambs per year. This biological difference in reproductive capacity between sexes makes male-focused selection both practical and effective for smallholder systems.

From lines 112 to 120, we added the following paragraph to the manuscript to further clarify why selection is performed on males.

“In the present study, the selection was carried out only on males, with no selection criteria imposed on females (apart from breeding soundness). This male-focused approach was adopted because implementing female selection is practically difficult in smallholder systems, where individual farmers typically maintain only a limited number of breeding ewes. Biologically, a single superior ram can service flocks from 3-4 farmers and produce numerous offspring annually, whereas each ewe typically produces only 1-2 lambs per year. Given this difference in reproductive capacity between sexes, sustaining the genetic improvement requires maintaining a larger population of females while selecting fewer higher-quality males, an approach that proves both practical and effective for smallholder production systems.”

2. Material and Methods

Consider adding a country map highlighting the study area, and also add an image of Farta Sheep breed

Authors’ Responses:

Dear reviewer #1,

As per the suggestion, we added Figure 1 to show major areas where Farta sheep are densely populated. In addition, we have shown the picture of Farta sheep in Figure 2.

3. Minor comments

There is some repetition and minor grammatical issues

Authors’ Responses:

Dear reviewer #1,

We removed those sentences and phrases that seem repetitive (for instance: lines 124, 125, 149, 165, 223, 224, 266) and revised grammatical errors.

4. Strength

The manuscript is well structured with a logical flow of ideas. Citations are properly used to support key points. Breeding schemes are clearly explained with practical considerations. It sets the context by highlighting the importance of livestock in Ethiopia and provides adequate background on Farta sheep breed.

Authors’ Responses:

Dear reviewer #1,

Thank you for highlighting the strengths of our manuscript. We belief that our research will contribute to the sustainable utilization of the local genetic resources.

III. Comments and questions from reviewer #2

1. materials and methods too elaborate and not information

method of analysis not clearly explained

Authors’ Responses:

Dear reviewer #2,

We followed a standard methodological approach, which included:

Describing the production system,

Stating assumptions about the population,

Defining the breeding objective, and

Presenting economic values along with the genetic and phenotypic parameters of breeding objective traits.

Under these subtopics, we provided all necessary details to simulate various breeding schemes. We believe our methodological steps are sufficient to allow study replication and verification of findings.

Regarding the “method of analysis”, we call it “prediction of selection response and rate of inbreeding”, as it is a common approach in simulation studies.

We used the SelAction computer program (Rutten et al., 2002) to simulate selection responses. Since we cited the simulation program, we believe that presenting detailed mathematical formulas and models used by the software to predict responses is unnecessary. However, we described the program’s basic features, including the types of outputs it generates.

2. Table 1. Genetic and phenotypic parameter estimates of the study traits need standard error and only few traits have been studied and the reason for the same

Authors’ Responses:

Dear reviewer #2,

As suggested, we have included the standard errors for genetic and phenotypic parameters of the study traits in Tabe 1.

Regarding the question of why “only few traits have been studied and the reason for the same”: As an entry point for Farta sheep genetic improvement, we focused on four key breeding objective traits prioritized by smallholder farmers. Given the limited infrastructure and financial resources typical of smallholder systems, the sustainability of genetic improvement depends on maintaining technical simplicity in breeding programs. We determined that expanding beyond four traits could introduce practical challenges, particularly regarding data collection. However, we acknowledge that breeding programs should remain adaptable over time. Future revisions could include implementing a two-tier breeding structure, replacing existing traits, or incorporating new traits as needed. We have included these considerations in our concluding recommendations.

In the present study "Genetic and Economic Efficiencies of Alternative Breeding Schemes for Improvement of Local Breeds in Low-input Production Systems: The Case of the Farta Sheep in Northwest Ethiopia. The manuscript investigates sustainable genetic improvement in low-input livestock systems, focusing on indigenous breed, Farta sheep. The research explores the efficiencies of three alternative breeding programs to identify appropriate schemes that can sustainably enhance sheep productivity within existing breeding practices and infrastructure.

Major comments:

3. A key aspect of the study involves deriving genetic and phenotypic parameters for body weights from Farta or crossbred sheep. However, it is important to note that parameters for lambing rate (NLW) and fertility were adapted from studies on non-Ethiopian breeds, which may not accurately represent Ethiopian indigenous sheep in low-input systems.

Authors’ Responses:

Dear reviewer #2,

Thank you for raising this important point. Yes, we used genetic parameters for NLW and fertility traits from published studies on non-Ethiopian sheep breeds. This approach was necessary because: (1) insufficient phenotype and pedigree data were available for these traits in our study population, and (2) no published parameter estimates exist specifically for Ethiopian indigenous sheep. As noted in lines 238-240, we clearly stated our use of parameters from other breeds while acknowledging their limitations. This is a well-established practice in breeding program simulations when breed-specific estimates are unavailable. Importantly, this limitation does not influence the relative comparison of the three breeding schemes, as all were evaluated using the same parameter estimates. However, we recognize that the realized genetic gains for these traits may differ from our predictions. We note that both traits typically show low heritability, which explains the lowest annual genetic progress observed in our simulations. We have carefully considered this aspect in our discussion of the findings, particularly regarding the expected pace of genetic improvement in smallholder breeding programs.

4. Scheme II is presented as a suitable option for the genetic improvement of indigenous sheep in low-input systems due to its genetic and economic benefits, as well as its operational feasibility. However, the study acknowledges that the operational feasibility from a smallholder farmer's perspective in the Ethiopian context is not extensively detailed. Challenges such as controlled mating and centralized ram selection are difficult to implement given traditional communal grazing and multi-sire mating practices prevalent in Ethiopia.

Authors’ Responses:

Dear reviewer #2,

We appreciate your valuable comments regarding the implementation challenges of community-based breeding programs under smallholder conditions. While the issues you raised “… controlled mating and centralized ram selection …communal grazing and multi-sire mating…” are indeed important constraints, we would like to clarify that these challenges apply equally to all three breeding schemes evaluated in our study. For our comparative analysis, we specifically focused on the technical and operational challenges that differ among the three schemes (details lines 149 -203), as these distinctions are particularly relevant to evaluate the alternative breeding schemes.

We assumed a reasonably controlled mating practice would be implemented consistently in all the three schemes. In addition, data recording on breeding objective traits is mandatory, while participatory ram selection is expected at the village level for all three schemes. Our simulated breeding schemes work under these assumptions. We believe that genetic improvement cannot come for free, rather, it fundamentally requires the fulfillment of certain baseline conditions. Thus, prior to implementation, participating farmers (those who organize themselves to run the breeding scheme) should fully understand these operational requirements and commit to fulfilling their respective responsibilities. Our study explicitly accounts for these essential operational requirements, which serve as the foundation for implementing any successful breeding program under smallholder conditions.

For instance, our evaluation compared three ram-to-ewe mating ratios (1:10 (P =20%), 1:20 (P=10%), and 1:40 (P=5%)) across all breeding schemes. The 1:10 ratio, equivalent to individual farmers maintaining their own breeding ram, offers the advantage of easier mating control in crop-livestock systems where flocks are typically kept separate. However, this scenario yielded relatively low annual genetic gains. In contrast, the 1:40 ratio demonstrated genetic gains twice those of the 1:10 ratio, but requires 3-4 farmers to share a single ram. The 1:40 mating ratio demands careful management to prevent multi-sire mating, particularly when using communal grazing areas or watering points shared with other flocks. Farmers must recognize that higher genetic gains necessitate greater organizational effort and strict mating control.

Looking ahead, we suggested eventually incorporating artificial insemination not only to overcome natural mating challenges but also accelerate genetic progress. This transition would be particularly valuable for maintaining genetic gains while addressing the practical constraints of communal management systems.

5. While farmers' trait preferences are mentioned, the breeding objective does not fully explore local definitions of "good mothering ability" and culturally significant traits such as tail type and color.

Authors’ Responses:

Dear reviewer #2,

Good mothering ability is the most highly valued trait for breeding ewes among smallholder farmers in our study area. However, as this trait cannot be directly measured, farmers characterize it through various behavioral and (re)production indicators. The definitions and descriptions

---

## [Editor Report · Decision Letter 1]

20 Aug 2025

Genetic and Economic Efficiencies of Alternative Breeding Schemes for Improvement of Local Breeds in Low-input Production Systems: The Case of the Farta Sheep in Northwest Ethiopia

PONE-D-25-28186R1

Dear Dr. Abebe,

We’re pleased to inform you that your manuscript has been judged scientifically suitable for publication and will be formally accepted for publication once it meets all outstanding technical requirements.

Kind regards,

Dawit Tesfaye, Prof.

Academic Editor

PLOS ONE

Additional Editor Comments (optional):

The authors have tried their best to address the issues raised during the first review process. The limitation of their studies are clearly stated especially the use of some phenotypic data form other non Ethiopian sheep breeds due to absence of such information for Farta sheep.
---

## [Editor Report · Acceptance letter]

PONE-D-25-28186R1

PLOS ONE

Dear Dr. Abebe,

I'm pleased to inform you that your manuscript has been deemed suitable for publication in PLOS ONE. Congratulations! Your manuscript is now being handed over to our production team.

Kind regards,

on behalf of

Dr. Dawit Tesfaye

Academic Editor

PLOS ONE